# Adulticidal and Repellent Activity of Essential Oils from Three Cultivated Aromatic Plants Against *Musca domestica* L.

**DOI:** 10.3390/insects16050542

**Published:** 2025-05-20

**Authors:** Gabriela Antonieta Oyarce, Patricia Loyola, Michelle Iubini-Aravena, Álvaro Romero, J. Concepción Rodríguez-Maciel, José Becerra, Gonzalo Silva-Aguayo

**Affiliations:** 1Facultad de Agronomía, Universidad de Concepción, Avenida Vicente Méndez 595, Chillán 3812120, Chile; 2Envu, Cary, NC 27513, USA; 3Posgrado en Fitosanidad-Entomología y Acarología, Colegio de Posgraduados, Carretera Federal Mexico Texcoco 36 5, Texcoco de Mora 56230, Mexico; 4Laboratorio de Química de Productos Naturales, Facultad de Ciencias Naturales y Oceanográficas, Universidad de Concepción, Víctor Lamas 1290, Concepción 4070386, Chile

**Keywords:** essential oils, biopesticides, pest management, house fly

## Abstract

House flies transmit diseases to both animals and humans. Their control relies on using insecticides, but these chemicals are becoming less effective and raise safety concerns. This study evaluated the use of essential oils from eucalyptus, fennel, and sage as natural fly control options. The laboratory results indicated that all three oils exhibited insecticidal and repellent properties. Eucalyptus oil was the most effective fumigant, killing all house flies at 34 µL L^−1^ of air, while fennel oil demonstrated the highest contact toxicity, achieving 100% mortality at 150 µL L^−1^. All oils repelled over 87% of flies. These findings underscore essential oils as eco-friendly alternatives to synthetic insecticides.

## 1. Introduction

The house fly, *Musca domestica* L. (Diptera: Muscidae), is an insect species closely linked to human settlements and livestock production systems. It has a cosmopolitan distribution and is commonly found in urban, peri-urban, and agricultural areas, especially in buildings that house birds and livestock [1]. This species rapidly completes its life cycle, often in less than two weeks, and can produce up to 12 generations in a single summer [2]. House flies flourish in environments abundant in organic matter, such as animal manure, soiled animal bedding, and household waste [3].

*M. domestica* is a pest because of its feeding and reproductive behaviors, and it acts as a vector for over 100 species of pathogenic microorganisms that affect humans and domestic animals. These pathogens include those responsible for infant diarrhea, anthrax, cholera, taeniasis, typhoid fever, trypanosomiasis, anaplasmosis, and tuberculosis, among others [4]. An increase in house fly populations causes animal discomfort and stress, reducing livestock product yields and complicating their commercialization when contamination occurs [3].

Synthetic contact insecticides have been used for years to control this pest due to their low cost and effectiveness [5]. However, the extensive use of these insecticides has resulted in resistance among house fly populations, primarily to pyrethroids, with over 80 documented cases worldwide [6]. Additionally, users are increasingly concerned about synthetic insecticides’ environmental risks and consequences [7]. Therefore, there is an urgent need to explore alternative control methods that are environmentally friendly, non-toxic, and less likely to cause resistance.

Among these alternatives are essential oils (EOs), complex aromatic substances derived from plants. These oils play vital roles in plant defense and signaling, such as attracting pollinators and beneficial insects [8,9]. The main components of EOs are plant secondary metabolites with significant pest control potential [10]. These are well established as multifunctional bioactive agents, with comprehensive reviews documenting their broad-spectrum insecticidal, repellent, and growth-disrupting properties against arthropod pests [11,12,13]. These studies highlight their efficacy in fumigation, oviposition deterrence, and contact toxicity, mechanisms often linked to their dominant terpenoid constituents. Botanical extracts are also environmentally friendly, safe, and non-toxic to humans, pets, and other non-target organisms [14]. Additionally, the large-scale production of plant EOs for the perfume and food industries ensures their availability at low prices, and their extraction does not harm the environment [15]. This research aimed to evaluate the fumigant properties, contact toxicity, and repellency of EOs from eucalyptus (*Eucalyptus globulus* Labill.), fennel (*Foeniculum vulgare* Mill.), and sage (*Salvia officinalis* L.) against adult *M. domestica* under laboratory conditions.

## 2. Materials and Methods

### 2.1. Chemicals and Reagents

Eucalyptus (*Eucalyptus globulus*; Product Code: 7545) and sage (*Salvia officinalis*; Product Code: 7610) essential oils were commercially sourced from Now^®^ Essential Oils (Now Foods, Bloomingdale, IL, USA), with certified 100% purity. According to the manufacturer’s specifications, both oils were distilled with steam, with eucalyptus distilled from leaves and small branches, and sage distilled from partially dried leaves. In contrast, fennel (*Foeniculum vulgare*) essential oil was produced in-house via hydro distillation using a Clevenger apparatus (4 h, 1:10 plant-to-water ratio) and dried aerial parts (leaves and flowers; harvested [March/2018]). For all bioassays, the essential oils were diluted in high-purity acetone (99.5%, Sigma-Aldrich, Schnelldorf, Germany; CAS 67-64-1) to ensure consistency in the test solutions.

### 2.2. Insect Rearing

Adult specimens of *M. domestica* were collected in the field using entomological nets at the Estación Experimental Ganadera of the Faculty of Veterinary Sciences at Universidad de Concepción and then transferred to the Laboratorio de Entomología y Acarología Agropecuaria at the Faculty of Agronomy of the Universidad de Concepción. Management practices for house flies at the experimental station included the removal of animal feces and bedding every 15 days, along with the application of AGITA^®^ WG (Neonicotinoid, Thiametoxam insecticide [10% (*w*/*w*), water-dispersible granules, Kwizda Agro, Meckesheim, Germany]) at the start of the summer season. The origin of the individuals used in the bioassays was not specified as they were not reared in discrete generations. To prevent inbreeding, field-collected individuals were added to the colony annually.

Rearing was performed based on the protocol described by [15], with modifications. *M. domestica* specimens were kept at 27 ± 1 °C, in 50 ± 5% relative humidity, and under a 16:12 h light/darkness photoperiod in a bioclimatic chamber (Memmert Gmbh IPS 749, Schwabach, Germany) inside cubic-shaped meshes (50 × 50 × 50 cm). Adults were fed ad libitum with distilled water and granulated sugar in separate Petri dishes. A wheat bran and whole milk mixture with 2%(*w*/*v*) granulated sugar was provided for oviposition. Eggs were collected twice a week and stored in modified paper containers. Larvae were fed with moistened strips of paper with milk and 2% (*w*/*v*) sugar disposed inside the containers.

### 2.3. Chemical Analysis of EOs

The chemical composition of EOs was determined using a gas chromatograph (GC, Agilent 7890, Santa Clara, CA, USA) coupled with an HP-5MS capillary column (length of 30 m, inner diameter of 0.25 mm, thin film of 0.25 µm and injection volume of 1 µL). The initial temperature of the equipment was 60 °C, which was maintained for 5 min, with an increase of 10 °C every minute until 280 °C, where it was held for 15 min. The relative percentages of the compounds from *E. globulus*, *F. vulgare* and *S. officinalis* were obtained using helium as the mobile phase and a mass selective detector (Agilent 5975 C, Santa Clara, CA, USA), and the components were identified using the NIST/EPA/NIH Mass Spectral Library (NIST 17).

### 2.4. Bioassays

#### 2.4.1. Fumigant Toxicity Assay

Fumigant activity was evaluated using a device developed by the Laboratorio de Entomología y Acarología Agropecuaria [16] (Figure 1). The setup consisted of a 500 mL plastic container with a modified lid containing two perforations. A small test tube lid was inserted through the perforations, and an Eppendorf tube was secured using a transparent tulle mesh. Inside the Eppendorf tube, a piece of filter paper was placed and impregnated with the corresponding volume of EOs. The tulle mesh acted as a physical barrier to prevent direct contact between the flies and the impregnated filter paper while allowing the dispersion of volatile compounds within the container. A cotton swab moistened with distilled water and a 2% (*w*/*v*) sugar solution was placed inside the container. Each treatment included five replicates and fifteen adult flies (1–3 days old) per container, without sex differentiation. The untreated control consisted of filter paper placed in the tube without impregnation. Mortality was assessed 48 h after exposure to the treatment.

#### 2.4.2. Contact Toxicity Assay

The insecticidal activity was assessed in adult flies using a modified version of the method described by [17], where individuals were exposed topically instead of to impregnated filter paper for better exposure control. Three-day-old flies, without sex differentiation, were immobilized by placing them at −20 °C for 90 s. Then, 1 µL of each essential oil (EO), diluted in acetone, was applied to the ventral side of the abdomen. Each treatment consisted of five replicates, with fifteen adult flies per replicate, and an untreated control (acetone only) was included. After treatment, the flies were transferred to 500 mL plastic containers containing a cotton ball saturated with distilled water and a 2% (*w*/*v*) sugar solution. The containers were covered with tulle netting. Mortality was recorded 24 h after treatment.

#### 2.4.3. Repellency Assay

Repellency was evaluated using the method of [18] with modifications (Figure 2). Two Büchner flasks were placed in an entomological cage (100 × 80 × 80 cm) and connected to an aquarium air pump probe. The airflow, previously filtered with activated carbon and distilled water, was displaced through the probe. Inside each flask, 1 mL of 0.5, 1.0, and 2.0% (*v*/*v*) solution of diluted EO was added to 9 mL of milk for the treatment and 1 mL of acetone with 9 mL of milk for the control. The airflow was constantly adjusted to 0.5 L min^−1^ with a flow meter. In the center of the cage, fifty 7-day-old flies were released, and the number of flies that landed in each flask was counted after 2 h. Five replicates were considered for each concentration, and the repellency percentage was calculated using the following formula [19]:$$R\left(\%\right)=\left[\frac{C\;-\;T}{C}\right]\;\times \;100$$

C = number of flies in the control, T = number of flies in the treatment.

**Figure 2 insects-16-00542-f002:**
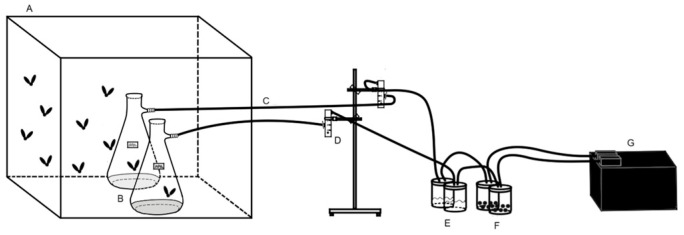
Scheme of the repellency test with the cage (A), the control and treatment Büchner flasks inside (B), the probe (C) that connects them to the flow meter (D), the flask with distilled water (E), the flask with activated carbon, (F) and the fish tank pump (G).

### 2.5. Data Analysis

Bioassays were conducted using a completely randomized experimental design. The data obtained were statistically analyzed using the Statistical Analysis System (SAS) software Version 9.00. [20]. Mortality percentages were corrected using Abbott’s formula [21], and mean comparisons were performed using a one-way analysis of variance (ANOVA), followed by Tukey’s test (*p* < 0.001). Additionally, a Probit analysis was conducted to estimate the LC_50_ and LC_90_ values with 95% confidence intervals (CI) using the R package “ecotox” [22]. LC values were considered significantly different when their 95% CIs did not overlap. Chi-square and *p*-values were also reported in this analysis.

## 3. Results

### 3.1. Chemical Composition of EOs

All EOs exhibited distinct chemical compositions, with a higher presence of monoterpene hydrocarbons (MHs). However, oxygenated monoterpenes (MOs) were the most abundant compounds (Figure 3). Eucalyptus and sage also contained sesquiterpene hydrocarbons (SHs) and oxygenated sesquiterpenes (SOs), with sage showing the most remarkable chemical diversity. In contrast, fennel was composed exclusively of monoterpenes.

Alpha-pinene was present in all EOs, with an abundance of at least 4%, while Beta-pinene, Beta-myrcene, and Gamma-terpinene each accounted for less than 1% of the total composition (Table 1). The predominant compounds in eucalyptus EO were the monoterpene Alpha-pinene (14%) and the MO 1,8-cineole (76.5%). In fennel, the MOs fenchone (29.5%) and trans-anethole (57%) were the main components, while in sage, the MOs camphor (18.8%) and Alpha-thujone (54.3%) were the most abundant.

### 3.2. Fumigant Toxicity

All the evaluated EOs showed fumigant toxicity against adult house flies (Figure 4). The three treatments (eucalyptus, fennel, and sage essential oils) achieved 100% mortality at 34, 50, and 50 µL L^−1^ air, respectively. At the LC_90_ level, eucalyptus essential oil proved to be the most effective treatment at 25.8 µL L^−1^ air. There were no significant differences in the LC_90_ between fennel and sage essential oils, with values of 41.2 and 47.7 µL L^−1^ air, respectively (Table 2).

### 3.3. Contact Toxicity

Similarly, all EOs exhibited contact toxicity (Figure 4). At the LC_90_ level, the treatment with the highest contact toxicity was fennel essential oil at 130 µL L ^−1^ (Table 3). The eucalyptus and sage essential oils behaved very similarly, as their cut-off values overlapped.

### 3.4. Repellency

All essential oils demonstrated repellency against adult *M. domestica*. Eucalyptus EO at a 5 µL mL^−1^ concentration showed significantly lower repellency (53%) than all other concentrations evaluated (Figure 5). In contrast, fennel and sage EOs exhibited repellency percentages more significant than 85% across all tested concentrations, with no significant differences.

## 4. Discussion

Monoterpenes are among the primary components of essential oils, consisting of two isoprene units formed by ten carbon atoms. They are divided into two main groups: hydrocarbon monoterpenes and oxygenated molecules. The latter are also known as monoterpenoids, and include oxygenated functional groups in their structure [9]. Recent studies indicate that the bioactivity of monoterpenes is linked to their effects on the nervous system, with the ability to alter the function of ion channels, such as γ-aminobutyric acid type A receptors (GABAARs), nicotinic acetylcholine receptors (nAChRs), tyramine receptors (TA), octopamine receptors (OA), transient receptor potential (TRP) channels, and enzymes like acetylcholinesterase (AChE) and Na^+^/K^+^-ATPase in insects [23,24].

Eucalyptus EO exhibited the strongest fumigant toxicity in our study, attributable to its main components, 1,8-cineole (76.5%) and α-pinene (14%). These results align with [25], which reported comparable fumigant activity for *Eucalyptus cinerea* F. Muell. ex Benth EO against *M. domestica* adults, achieving 100% mortality within 15 min. Notably, *E. cinerea* EO contained an even higher 1,8-cineole concentration (>90%), further supporting its pivotal role in toxicity. The insecticidal potency of 1,8-cineole is further supported by [26], where 1,8-cineole exhibited multiple toxic effects at 5.0 mg kg^−1^, causing direct larval toxicity through ingestion (LC_50_ = 1.59 mg kg^−1^), reducing pupation rates by 67.5% and adult emergence by 75%. Additionally, biochemical analyses revealed its capacity to competitively inhibit cytochrome P450 enzymes. These findings underscore 1,8-cineole’s efficacy as both a fumigant on adults and larvicide, highlighting its versatility as an insecticide.

The EO with the highest contact toxicity, *Foeniculum vulgare*, typically contains trans-anethole (48–90%) and fenchone (3–10%) as major components, along with minor constituents such as estragole, α-pinene, and p-cymene [12,27]. Our analysis revealed a similar profile, though with a notably higher fenchone content (29.5%) compared to the literature, while trans-anethole (57.3%) fell within the reported range (Table 1). The insecticidal activity of fennel EO and its primary components has been well documented. For example, [14] reported that trans-anethole exhibited the highest toxicity against *M. domestica* larvae and pupae (LC_50_ = 0.58%), followed by fennel EO (LC_50_ = 1.57%) and fenchone (LC_50_ = 17.22%). In addition, 100% larval mortality was achieved at 2.5% for trans-anethole and 10% for fennel EO. These findings align with [28], where it was demonstrated that trans-anethole and lemongrass EO (*Cymbopogon citratus*; 45.23% geranial) showed superior adulticidal activity compared to α-cypermethrin (1%). Notably, their combination caused 100% mortality in under 5 min (KT_50_ = 3.2 min) and induced morphological deformities in larval mouthparts and antennae. Importantly, these compounds exhibited low toxicity to non-target organisms (*Tetragonula pegdeni* and *Poecilia reticulata*). Collectively, these results highlight trans-anethole as the primary driver of fennel EO’s contact toxicity, particularly against *M. domestica* larvae and pupae. Moreover, its efficacy can be significantly enhanced through synergistic combinations with other bioactive compounds, suggesting promising potential for targeted formulations.

While *Salvia officinalis* EO exhibits a variable composition [29,30,31,32], it is typically dominated by camphor, α-thujone, 1,8-cineole, viridiflorol, and β-thujone. Our chemical analysis revealed a profile particularly rich in α-thujone (54.3%) and camphor (18.8%), with minimal 1,8-cineole (2.2%). These findings correlate well with the bioactivities reported by [29], where both sage and rosemary (*Rosmarinus officinalis*) EOs demonstrated strong repellency and oviposition inhibition against *Calliphora vomitoria* (2.5 μL/cm^2^), along with notable contact toxicity (LD_50_ = 1.2 μL/insect) and fumigant activity (LC_50_ = 16.64 µL L^−1^ air).

The insecticidal properties of these major components have been further characterized in other diptera species. For example, [33] demonstrated that thujone exhibits broad-spectrum toxicity against *Bactrocera dorsalis* (Diptera: Tephritidae), affecting all life stages (LC_50_ < 66 mg/mL) and achieving complete oviposition deterrence at 5% (*v*/*v*). Conversely [34], found camphor to be relatively ineffective against *M. domestica*, showing <50% repellency and low toxicity even in susceptible strains. Notably, the repellent activity observed in our study appears primarily attributable to α-thujone, given its high concentration and the minimal contribution of 1,8-cineole. This aligns with established mechanisms of monoterpene activity, where compounds like thujone are known to interact with insect odorant-binding proteins, disrupting normal behavioral responses [35].

Essential oils and their components can effectively control agricultural pests. Field studies conducted in the USA, Chile, and other countries show that EO-based insecticides effectively manage soft-bodied insects, stinging insects, and mites [15]. These products can be used alone or in combination with conventional insecticides, enhancing their toxicity through synergistic interactions [36]. Similar effects have been observed in insect pests exposed to pyrethroids, such as permethrin, deltamethrin, β-cyfluthrin, and natural pyrethrins [36,37]. Furthermore, insecticides from various chemical classes have produced similar outcomes, including imidacloprid [38], temephos, malathion [39], phosphine gas [40], and neem-oil-based bioinsecticides [41]. Combining conventional insecticides with EOs may help reduce the chemical burden on the environment and lessen the risk of resistance developing in pest populations [42].

However, their commercial use requires careful consideration of potential irritant effects on mammals. Key components such as α-thujone (neurotoxicity, dermatitis), camphor (respiratory irritation), and 1,8-cineole (bronchial spasms) pose documented risks to airways and skin at high concentrations [43,44,45,46]. As demonstrated by formulated products like cinnamon-EO-based bioherbicides (patent WO 2019/238948) [47], proper dilution and encapsulation can reduce toxicity risks by over 100-fold while maintaining efficacy [48]. This highlights the critical next step for our findings: developing optimized formulations of the tested EOs (eucalyptus, fennel, sage) to maximize their pest control potential while minimizing mammalian exposure risks.

## 5. Conclusions

All the tested essential oils demonstrated significant topical and fumigant toxicity and strong repellency against adult *M. domestica*. The evidence indicates that these activities are closely associated with the most abundant compounds found in essential oils. However, it is vital to confirm these findings by assessing the toxicity of these individual compounds on *M. domestica*. The results emphasize the potential ability of essential oils and their components to control *M. domestica*, and could be incorporated into pest management programs or used to develop new insecticidal products. These products could also be utilized against other dipteran pests of agricultural and medical significance.

## Figures and Tables

**Figure 1 insects-16-00542-f001:**
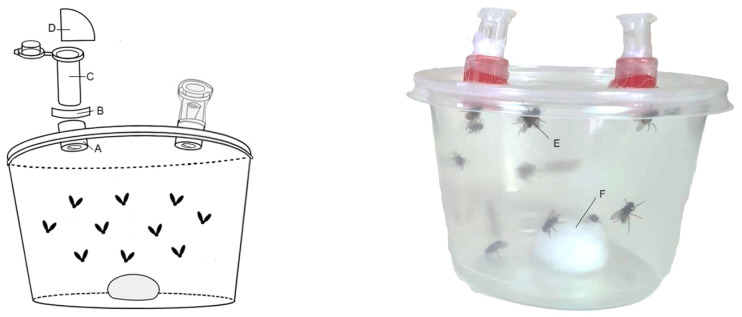
Outline of the components of the fumigant test device, showing the sample test tube lid (A), the tulle mesh (B), the Eppendorf tube (C), the filter paper (D), adult flies (E), and moistened cotton swab (F).

**Figure 3 insects-16-00542-f003:**
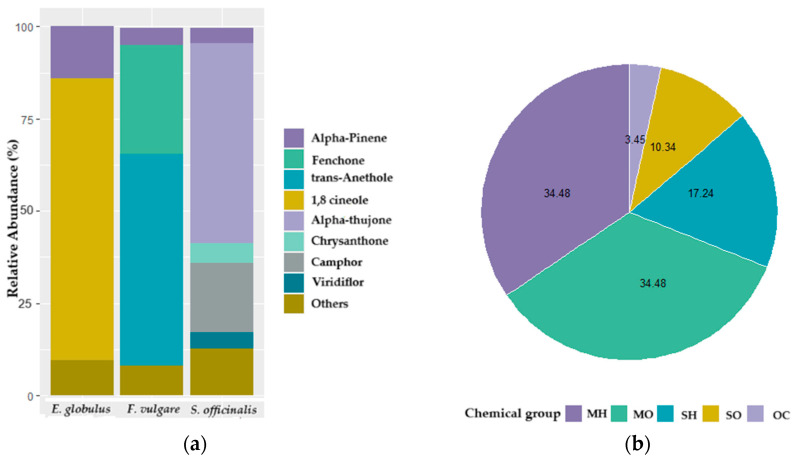
Chemical composition of eucalyptus, fennel, and sage essential oils. (**a**) A stacked plot bar shows each tested EO’s relative abundance of components. “Others” represents the sum of 35 compounds with an overall abundance lower than 5%. (**b**) Circular plot shows the percent of compounds, and their chemical classes identified in all the EOs studied (MH: Monoterpene hydrocarbon, MO: Monoterpene oxygenated, SH: Sesquiterpene hydrocarbon, SO: Sesquiterpene oxygenated, OCs: other compounds).

**Figure 4 insects-16-00542-f004:**
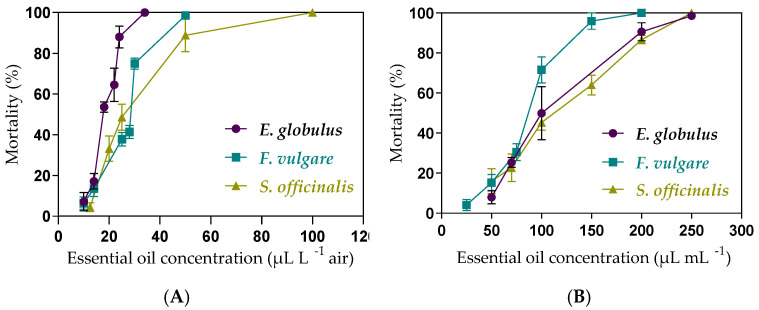
(**A**) Fumigant and (**B**) contact toxicity of the EOs of eucalyptus, fennel, and sage against *M. domestica* adults.

**Figure 5 insects-16-00542-f005:**
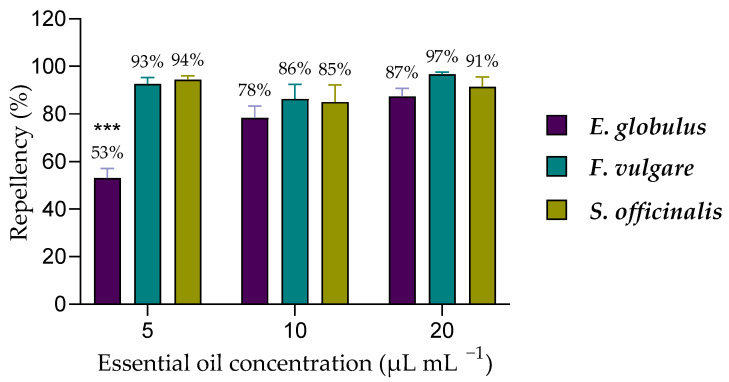
Repellence of *M. domestica* by eucalyptus, fennel, and sage EOs. “***” indicates significant differences in percent repellency between concentrations as determined by a one-way ANOVA test with a *p*-value (<0.005).

**Table 1 insects-16-00542-t001:** Chemical composition, relative abundance, and classification of the components from the evaluated essential oils.

Compounds	RT (min) ^1^	*Eucalyptus* *globulus* (%)	*Foeniculum* *vulgare* (%)	*Salvia* *officinalis* (%)
α-Pinene	5.702	14	4.644	4.03
Camphene	5.976	-	0.366	2.39
β-Phellandrene	6.455	-	0.108	-
β-Pinene	6.517	0.312	0.257	0.437
β-Myrcene	6.79	0.364	0.556	0.424
α-phellandrene	7.056	-	1.759	-
o-Cymene	7.449	-	0.143	0.277
D-limonene	7.541		1.897	0.916
1,8-Cineole	7.579	76.475	-	2.16
Γ-terpinene	8.127	0.272	0.438	0.27
L-Fenchone	8.75	-	29.489	-
α-Thujone	9.125	-	-	54.32
Chrysanthone	9.27	-	-	5.33
Camphor	9.821	-	0.466	18.8
Borneol	10.189	-	-	1.052
Terpinen-4-ol	10.403	-	-	0.275
Estragole	10.766	-	2.14	-
Bornyl acetate	12.331	-	-	0.565
Trans-anethole	12.434	-	57.278	-
Camphene	13.405	3.07	-	-
α-Gurjunene	14.45	2.589	-	-
Caryophyllene	14.613	-	-	1.583
Humulene	15.151	-	-	1.352
Aromadendrene	15.264	1.389	-	-
(+)-Ledene	15.79	-	-	0.51
Epiglobulol	16.769	0.204	-	-
Globulol	17.137	0.966	-	-
Viridiflorol	17.251	0.358	-	4.21
Epimanool	24.027	-	-	0.588
Monoterpene hydrocarbons		18.018 (5)	10.168 (9)	8.744 (7)
Oxygenated monoterpenes		76.475 (1)	89.373 (4)	82.502 (7)
Sesquiterpene hydrocarbons		3.978 (2)	**-**	3.445 (3)
Oxygenated sesquiterpenes		1.528 (3)	**-**	4.21 (1)
Other compounds		-	**-**	0.588 (1)
Total identified (%)		99.999	99.541	93.489

^1^ RT = Retention index.

**Table 2 insects-16-00542-t002:** Fumigant toxicity for lethal concentrations of 50% (LC_50_) and 90% (LC_90_) of the evaluated essential oils against *M. domestica* adults.

Essential Oil	LC_50_ (LC 95%) * (µL L^−1^ air)	LC_90_ (LC 95%) * (µL L^−1^ air)	Equation	Chi-Square	PGOF **
Eucalyptus (*Eucaliptus globulus*)	18.1 a [17.3–19.0]	25.8 a [24.4–27.6]	Y = 0.168x − 3.05	6.47	0.167
Fennel (*Foeniculum vulgare*)	26.6 b [22.3–31.9]	41.2 b [34.9–56.8]	Y = 0.088x − 2.34	13.8	0.008
Sage (*Salvia officinalis*)	28.2 b [20.8–39.9]	47.7 b [37.2–85.4]	Y = 0.0658x − 1.85	9.25	0.02

* Lethal concentrations at 50 and 90% effectiveness with their respective fiducial limits at 95% probability, ** *p*-values for the goodness-of-fit test. Different letters in the same column indicate that their confidential limits do not overlap.

**Table 3 insects-16-00542-t003:** Contact toxicity of essential oils at lethal concentrations of 50% (LC50) and 90% (LC90) against adult Musca domestica.

Essential Oil	LC_50_ (LC 95%) * (µL mL^−1^)	LC_90_ (LC 95%) * (µL mL^−1^)	Equation	Chi-Square	PGOF **
Eucalyptus (*Eucaliptus globulus*)	111 ab [91.1–131]	180 b [155–229]	Y = 0.0184x − 2.03	10.5	0.033
Fennel (*Foeniculum vulgare*)	86.7 a [81.2–92.5]	130 a [121–142]	Y = 0.0297x − 2.57	3.51	0.476
Sage (*Salvia officinalis*)	117 b [108–127]	203 b [188–222]	Y = 0.0150X − 1.76	4.15	0.386

* Lethal concentrations at 50 and 90% effectiveness with their respective fiducial limits at 95% probability, ** *p*-values for the goodness-of-fit test. Different letters in the same column indicate that their confidential limits do not overlap.

## Data Availability

Raw data are not publicly available but may be obtained upon request.

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
