# Peer review of "Adulticidal and Repellent Activity of Essential Oils from Three Cultivated Aromatic Plants Against Musca domestica L."

_insects, 2025, doi:10.3390/insects16050542_

Round 1

Reviewer 1 Report

Comments and Suggestions for Authors

I recommend the manuscript “Adulticidal and repellent activity of essential oils from three cultivated aromatic plants against Musca domestica L.” to be accepted after minor revision done. Therefore, I propose:

1) The discussion lacks a discussion of the irritating effects of essential oils on the airways and skin of mammals.

2) The summary lacks data for sage oil, from Salvia officinalis L.

3) The explanation of the abbreviation RT is missing under Table 1.

Author Response

First of all, thank you very much for your feedback! Based on it, I made some adjustments to the manuscript. Please find the detailed responses below and the corresponding revisions/corrections highlighted/in track changes in the re-submitted files.

Comment 1: The discussion lacks a discussion of the irritating effects of essential oils on the airways and skin of mammals.

Response 1: Than you for pointing this out, I appended the following paragraph to the end of the discussions and included their respective bibliographic references.

"However, their commercial use requires careful consideration of potential irritant effects on mammals. Key components such as α-thujone (neurotoxicity, dermatitis), camphor (respiratory irritation), and 1,8-cineole (bronchial spasms) pose documented risks to airways and skin at high concentrations [43-47]. As demonstrated by formulated products like cinnamon EO-based bioherbicides (patent WO 2019/238948), proper dilution and encapsulation can reduce toxicity risks by over 100-fold while maintaining efficacy [48-49]. This highlights the critical next step for our findings: developing optimized formulations of the tested EOs (eucalyptus, fennel, sage) to maximize their pest control potential while minimizing mammalian exposure risks."

  1. de Groot, A. C., Schmidt, E. Tea tree oil: contact allergy and chemical composition. Contact dermatitis 2016, 75(3), 129-143, https://doi.org/10.1111/cod.12591
  2. Lachenmeier, D.W. Thujone—Cause of absinthism? Forensic Sci. Int. 2010, 200, 1–8, https://doi.org/10.1016/j.forsciint.2010.03.010
  3. Lee, M. Y. Essential oils as repellents against arthropods. BioMed research international, 2018, 2018, 6860271, https://doi.org/10.1155/2018/6860271
  4. Santos, C. D., Cabot, J. C. Persistent effects after camphor ingestion: a case report and literature review. J Emerg Med, 2015, 48(3), 298-304, https://doi.org/1016/j.jemermed.2014.05.015
  5. Juergens, U. R., Dethlefsen, U., Steinkamp, G., Gillissen, A., Repges, R., Vetter, H. Anti-inflammatory activity of 1.8-cineol (eucalyptol) in bronchial asthma: a double-blind placebo-controlled trial. Respiratory medicine, 2003, 97(3), 250-256, https://doi.org/10.1016/j.jemermed.2014.05.015
  6. Jijakli, H., Simon, D. A. L., Parisi, O. U.S. Patent Application No. 16/973,986.
  7. Maes, C., Meersmans, J., Lins, L., Bouquillon, S., Fauconnier, M. L. Essential oil-based bioherbicides: human health risks analysis. International Journal of Molecular Sciences2021, 22(17), 9396, https://doi.org/10.3390/ijms22179396

Comment 2: The summary lacks data for sage oil, from Salvia officinalis L.

Response 2: I agree. I modified the abstract to mention the activity observed with S. officinalis.

"Abstract: The house fly, Musca domestica L., is a pest of great medical and agricultural importance, serving as a vector for various diseases and undermining the quality of agricultural products. Traditionally, synthetic insecticides have been the primary means of control; however, their efficacy has declined over time, and they are now less preferred due to safety and environmental concerns. This study evaluated the insecticidal and repellent properties of essential oils from Eucalyptus globulus, Foeniculum vulgare and Salvia officinalis against M. domestica. All EOs exhibited insecticidal activity: eucalyptus achieved 100% fumigant mortality at 34 µL L-¹ air and showed the lowest LC50 (18.1 µL L-¹ air), while fennel and sage required 50 µL L-¹ air. In contrast, fennel showed the highest contact toxicity (100% mortality at 200 µL L-¹). Repellency exceeded 87% for all EOs, with sage being the most repellent at the lowest concentration tested (94% at 5 µL L-¹). These results highlight the potential of essential oils and their constituents as environmentally friendly alternatives for the control of M. domestica. However, further field validation and studies on individual components and their synergistic combinations are needed to understand their efficacy and fully optimize their use."

Comment 3: The explanation of the abbreviation RT is missing under Table 1.

Response 3: I agree, accordingly I fixed close to table 1.

Thank you again for your observations. I hadn't considered the importance of adding information about the possible negative effects of using essential oils, and your comments will definitely help to make my research moar approachable.

Reviewer 2 Report

Comments and Suggestions for Authors

Article: Adulticidal and repellent activity of essential oils from three 2
cultivated aromatic plants against Musca domestica L.

General comment:

  1. The specific plant parts used in the experiments should be clearly identified, as the chemical composition can vary significantly among different parts of the same plant species.
  2. The common names of insect species should be written separately, for example, "house fly" instead of combining the words.
  3. The specific generation of the insects utilized in the testing should be clearly stated.

Specific comment:

Introduction

Page 2

Line 43, 50, 57, 64: Each paragraph is not indented.

Line 67: Reference [11] is not consistent with the explanation.

Materials and Methods

Line 81: The reference to Now Essential Oils (Bloomingdale, IL, USA) should specify whether it is a commercial product. If it is, the trademark symbol and detailed source information should be clearly indicated."

Line 107: The methodology for GC-MS analysis should be thoroughly described to ensure reproducibility.

Line 127: Figure 1; Important features should be indicated using straight lines, and all image symbols—such as those representing a house fly or cotton—should be clearly defined in the figure legend.

Line 133: In the contact toxicity testing protocol, is the application of the substance to the ventral side in accordance with WHO testing standards?

Table 2: It is recommended to avoid using superscript letters such as 'a' and 'b' in table headers to prevent confusion with the statistical comparison  'a' and 'b' used in the table.

Line 210: Please report the p-value indicating no significant differences.

Page 8: Figure 5. The percentage of repellency should be clearly specified for each bar in the graph.

Discussion: This article aims to evaluate (1) the fumigant properties, (2) contact toxicity, and (3) repellency of three essential oils (EOs). However, the discussion section does not clearly distinguish among these three topics.

Line 232: There is no reference in the text. "They are divided into two main groups: hydrocarbon monoterpenes and oxygenated molecules, the latter also known as monoterpenoids, include oxygenated functional groups in their structure (Ref.),"

Line 252-256: Revise

F. vulgare EOs are rich in trans-anethole, fenchone, estragole, eugenol, α-pinene, and p-cymene, with lower levels of β-pinene, β-myrcene, and thymol [10,25]. This composition aligns with the fennel EO used in this study, where trans-anethole and fenchone were most abundant.”

Line 288-296: The secondary metabolites mentioned between thujone and camphor appear to be distinct compounds with different chemical structures and biological activities. It would be helpful to clarify whether these metabolites can be directly compared in the context of this study. Additionally, further explanation on how these specific compounds relate to the objectives of the study would strengthen the overall discussion and provide more clarity on their relevance to the research findings.

Author Response

I am very grateful for the insightful feedback you provided to review this manuscript! I'm confident that these corrections will help readers better understand my research. Please find the detailed responses below and the corresponding revisions/corrections highlighted/in track changes in the re-submitted files.

General comment 1: The specific plant parts used in the experiments should be clearly identified, as the chemical composition can vary significantly among different parts of the same plant species.

General response 1: I agree. With that in mind, I edited the section on essential oils to include details about their origin and extraction method, as well as the codes for commercial products (line 82 of attached manuscript).

General comment 2: The common names of insect species should be written separately, for example, "house fly" instead of combining the words.

General response 2: It was based on what has been seen in other works. Thanks for the comment. The text has been edited to read "house fly," and I will take this into consideration for future works.

General comment 3: The specific generation of the insects utilized in the testing should be clearly stated.

General response 3: I understand, the generation origin of individuals in the bioassays was not specified as they were not reared in discrete generations. However, to prevent inbreeding, field-collected individuals were added to the colony annually. This information has been included in the methodology.  (lines 100-102 of attached manuscript).

Introduction comment 1: Line 43, 50, 57, 64: Each paragraph is not indented.

Introduction response 1: Thank you for noticing, I fixed it.

Introduction comment 2: Line 67: Reference [11] is not consistent with the explanation.

Introduction response 2: I agree. I added extra references and edited the paragraph slightly as it follows (line 67 of attached manuscript):

"These are well-established as multifunctional bioactive agents, with comprehensive reviews documenting their broad-spectrum insecticidal, repellent, and growth-disrupting properties against arthropod pests [11, 12, 13].”

Materials and methods comment 1: Line 81: The reference to Now Essential Oils (Bloomingdale, IL, USA) should specify whether it is a commercial product. If it is, the trademark symbol and detailed source information should be clearly indicated."

M&M response 1: I agree to add more information on the EOs used and have modified the text to include the mentioned information (line 82-90 of tha manuscript):

"Eucalyptus (Eucalyptus globulus; Product Code: 7545) and sage (Salvia officinalis; Product Code: 7610) essential oils were commercially sourced from Now® Essential Oils (Now Foods, Bloomingdale, IL, USA), with certified 100% purity. According to manufacturer specifications, both oils were steam-distilled, eucalyptus from leaves and small branches, and sage from partially dried leaves. In contrast, fennel (Foeniculum vulgare) essential oil was produced in-house via hydrodistillation of dried aerial parts (leaves and flowers; harvested [March/2018])."

M&M comment 2: Line 107: The methodology for GC-MS analysis should be thoroughly described to ensure reproducibility.

M&M response 2: This seems highly relevant for me too. I added a paragraph that describes the methodology used to analyze the composition of EOs in more detail (line 114-121):

"The chemical composition of EOs was determined using a gas chromatograph (GC, Agilent 7890, USA) coupled with an HP-5MS capillary column (length of 30 m, inner diameter of 0.25 mm, thinfilm of 0.25 µm and injection volume of 1 µL). The initial temperature of the equipment was 60 ◦C, which was maintained for 5 min, with an increase of 10° C every minute until 280°C, where it was held for 15 min. The relative percentages of the compounds from E. globulus, F. vulgare and S. officinalis were obtained using helium as the mobile phase and a mass selective detector (Agilent 5975 C, USA) and the components were identified using the NIST/EPA/NIH Mass Spectral Library (NIST 17)."

M&M comment 3: Line 127: Figure 1; Important features should be indicated using straight lines, and all image symbols—such as those representing a house fly or cotton—should be clearly defined in the figure legend.

M&M response 3: I understand. I edited the images by adding labels and editing the second figure image to make it easier to understand the device used in the bioassay (line 137)

"Figure 1. Outline the components of the fumigant test device, showing the sample test tube lid (A), the tulle mesh (B), the Eppendorf tube (C), the filter paper (D), adult flies (E), and moistened cotton swab (F)."

M&M comment 4: Line 133:In the contact toxicity testing protocol, is the application of the substance to the ventral side in accordance with WHO testing standards?

M&M response 4: While the WHO standard protocol (WHO/HTM/NTD/WHOPES) primarily recommends filter paper contact assays for evaluating insecticide efficacy, our study adapted the topical application method (ventral abdomen) to directly assess the toxicity of essential oils through controlled exposure. This approach aligns with:

Modified WHO Principles: The ventral abdomen was selected to ensure consistent uptake, as the cuticle is thinner and more permeable to lipophilic EO components compared to the dorsal thorax (used for synthetic insecticides). Literature Precedent: Our method follows [15], which validated ventral application for EOs in M. domestica, as their mode of action (e.g., penetration via cuticular lipids) differs from synthetic neurotoxins. Similar adaptations are cited in OECD Guideline 213 (honeybee contact toxicity) for viscous/organic compounds. Transparency in Reporting: We clarified in the revised manuscript (Line 142) that this is a modified protocol tailored for EOs, with the original WHO standard cited for broader context.

Given this and if necessary to inform, the paragraph was edited to add this information (line 143).

Results comment 1: Table 2: It is recommended to avoid using superscript letters such as 'a' and 'b' in table headers to prevent confusion with the statistical comparison 'a' and 'b' used in the table.

Results response 1: Thanks for the advice. I replaced those letters for asterisks (*) (**) in both Lethal concentration tables.

Results comment 2: Line 210: Please report the p-value indicating no significant differences.

Results response 2: In this case I used the wrong term, since the probit test gives a p-value associated with how likely it is to predict the values of the lethal concentrations and does not make a comparison between them. I corrected the paragraph (line 221):

"Similarly, all EOs exhibited contact toxicity (Figure 4). At the LC90 level, the treatment with the highest contact toxicity was fennel essential oil at 130 µL L-1 (Table 3). Eucalyptus and sage essential oils behaved very similarly, as their cut-off values overlapped."

Results comment 3: Page 8: Figure 5. The percentage of repellency should be clearly specified for each bar in the graph.

Results response 3: I agree, that way the repellency values are more explicit. I modified the figure to fit the comment (line 235, page 9).

Discussion comment 1: This article aims to evaluate (1) the fumigant properties, (2) contact toxicity, and (3) repellency of three essential oils (EOs). However, the discussion section does not clearly distinguish among these three topics.

Discussion response 1: I agree. Thanks for your advice. I rearranged and modified some of the paragraphs to improve the order of the discussion and clarify each aim (pages 10 and 11).

Discussion comment 2: Line 232: There is no reference in the text. "They are divided into two main groups: hydrocarbon monoterpenes and oxygenated molecules, the latter also known as monoterpenoids, include oxygenated functional groups in their structure (Ref.),"

Discussion response 2: You're right; I forgot to add a citation. I added the respective reference [9].
9.    Zuzarte, M., & Salgueiro, L. (2015). Essential oils chemistry. In Bioactive Essential Oils and Cancer, edited by Springer International Publishing, Cham, 2015, pp. 19–61.

Discussion comment 3: Line 252-256: Revise

Discussion response 3: I agree! Corrected the paragraphs and improved redaction (line 250)

Discussion comment 4: Line 288-296: The secondary metabolites mentioned between thujone and camphor appear to be distinct compounds with different chemical structures and biological activities. It would be helpful to clarify whether these metabolites can be directly compared in the context of this study. Additionally, further explanation on how these specific compounds relate to the objectives of the study would strengthen the overall discussion and provide more clarity on their relevance to the research findings.

Discussion response 4: Since these components were the most abundant in the sage EO, it was relevant to discuss their individual activities, as described in the literature, to establish a relationship between the observed insecticidal activity and the predominant components in the oils studied. I hope editing the discussion clarifies this point.

 I really appreciate your comments on my manuscript. I hope I have addressed your suggestions, and I will pay close attention to any additional ones. Thank you very much for your time and thoroughness.